# Study on the Stability of Functionally Graded Simply Supported Fluid-Conveying Microtube under Multi-Physical Fields

**DOI:** 10.3390/mi13060895

**Published:** 2022-06-03

**Authors:** Tao Ma, Anle Mu

**Affiliations:** 1School of Mechanical and Precision Instrument Engineering, Xi’an University of Technology, Xi’an 710048, China; maluyao@haue.edu.cn; 2School of Mechanical Engineering, Henan University of Engineering, Zhengzhou 451191, China

**Keywords:** strain gradient theory, functionally graded material, multi-physical fields, size effect, critical velocity

## Abstract

The stability of functionally graded simply supported fluid-conveying microtubes under multiple physical fields was studied in this article. The strain energy of the fluid-conveying microtubes was determined based on strain gradient theory, and the governing equation of the functionally graded, simply supported, fluid-conveying microtube was established using Hamilton’s principle. The Galerkin method was used to solve the governing equation, and the effects of the dimensionless microscale parameters, temperature difference, and magnetic field intensity on the stability of the microtube were investigated. The results showed that the dimensionless microscale parameters have a significant impact on the stability of the microtube. The smaller the dimensionless microscale parameters were, the stronger the microscale effect of the material and the better the microtube stability became. The increase in the temperature difference decreased the eigenfrequency and critical velocity of the microtube and reduced the microtube stability. However, the magnetic field had the opposite effect. The greater the magnetic field intensity was, the greater the eigenfrequency and critical velocity were, and the more stable the microtube became.

## 1. Introduction

Microtubes have been widely used in micro-electro-mechanical systems [1], bioengineering [2], modern medicine [3], and other fields [4] for resonators, actuators, fluid transport, fluid storage, and drug delivery. As important functional components, the stability of microtubes has an important impact on the whole system. Therefore, it is important to further understand the dynamic behaviors of microtubes for their design and application.

Experiments have proven that the mechanical properties of materials at the microscale are completely different from those at the macroscale. The size effect plays a key role in microstructures [5,6,7], but classical continuum mechanics theory cannot accurately describe the dynamic behaviors of microtubes. Hence, to accurately design and fabricate a microstructure, several higher-order continuum mechanics theories have been introduced, such as modified couple stress theory, strain gradient theory, nonlocal continuum mechanics theory, and nonlocal strain gradient theory. These theories contain at least one microscale parameter, which can accurately describe the influence of the size effect on the microstructure performance.

Based on these higher-order continuum mechanics theories, some scholars have conducted considerable theoretical research on microtubes. The complex viscoelastically coupled nonlinear equations of a fluid-conveying microtube were presented by Ghayesh et al. [8] to analyze the influence of the velocity of the flowing fluid on the system dynamics. Guo et al. [9] established a three-dimensional (3D) theoretical model with modified coupled stress theory to study the effect of small length scales on two types of periodic motions. Dehrouyeh-Semnani et al. [10,11] investigated nonlinear size-dependent resonant characteristics for conveying fluid via extensible microtubes subjected to harmonic loads. Hu et al. [12] developed the nonlinear equations of motion for cantilevered, fluid-conveying microtubes to analyze the possible size-dependent nonlinear responses based on modified coupled stress theory. Based on strain gradient theory, Hosseini et al. [13] established a cantilever dynamics equation for fluid-conveying microtubes based on three length scale parameters. The effects of the length scale parameters, outer diameter, and length–diameter ratio on the critical velocity and natural frequency of flutter were investigated. Yang et al. [14] studied the static post-buckling problem and the size-dependent post-buckling configurations, and they analyzed the influence of the material length scale parameters, outer diameter, flow velocity, and Poisson’s ratio on the dynamic characteristics. Considering the size effects of the microflow and microstructure, Wang et al. [15] studied the in-plane and out-of-plane bending vibrations of end-clamped, fluid-conveying microtubes. The studies described above indicated that the length scale parameters of materials have a significant impact on the stability of fluid-conveying microtubes. However, as the application environments of microtubes become more and more complex and changeable, such as under the action of ultra-high temperatures, ultra-low temperatures, high magnetism, and other physical fields, homogeneous microtubules of a single material can no longer meet the application requirements.

To improve the stability of microtubes in complex environments, novel functionally graded materials (FGMs) have been proposed for microtubes. However, most research is currently focused on functionally graded nanotubes. For instance, Zhang et al. [16] employed the isogeometric finite element method to analyze the effects of boundary conditions, geometric properties, and material parameters on the frequencies of carbon-nanotube-reinforced and FGM-sector cylindrical shells. Based on non-local elasticity theory, Hołubowski et al. [17] studied the transverse vibrations of single-walled carbon nanotubes under a random load action. A nonlocal strain gradient elasticity approach was proposed by Farajpour et al. [18] to study the mechanical behaviors of fluid-conveying nanotubes. The effects of different nano-system/fluid parameters, including the fluid-solid interface and the flow speed, on the nonlinear resonance, were analyzed. Bahaadini et al. [19,20] used an extended Hamilton’s principle to obtain the size-dependent governing equations and associated boundary conditions, and they analyzed the free vibrations of a nanotube conveying nanoflow. Deng et al. [21] studied the size-dependent vibrations and stability of multi-span, viscoelastic, functionally graded, fluid-conveying nanotubes using a hybrid method.

The above studies indicated that the stability of functionally graded nanotubes can be adjusted by designing the volume fraction index. Therefore, owing to their excellent physical properties, functionally graded carbon nanotubes can be used in multiple physical fields, such as thermo-magnetic fields. Tong et al. [22] established the dynamics equation for functionally graded cantilevered nanotubes under thermo-magnetic coupling. The influences of the external magnetic field and temperature on the stability of functionally graded material nanotubes were investigated. Based on the nonlocal Euler–Bernoulli beam theory, Lyu et al. [23] derived the nonlinear governing equation of fluid-conveying carbon nanotubes in elastic media under the action of thermal and magnetic fields. The effects of the small scale, Knudsen number, elastic medium, magnetic field parameters, and temperature variations on the stability of the system were studied. Ghane et al. [24] analyzed the influences of nonlocal parameters, the strain gradient length scales, the magnetic nanoflow, the longitudinal magnetic field, and the Knudsen number on the eigenvalues and critical flutter velocity of fluid-conveying thin-walled nanotubes under the action of a magnetic field. Based on nonlocal elasticity theory and Euler–Bernoulli beam theory, Zhu et al. [25] established the governing equation of cantilevered carbon nanotubes subjected to partially distributed tangential forces, and they analyzed the influence of non-local parameters, the Knudsen number, surface effects, and the magnetic field on the stability of cantilevered carbon nanotubes. Bahaadini et al. [26] analyzed the stability of cantilevered carbon nanotubes subjected to axial compressive loads. Hosseini et al. [27,28] studied the stability of carbon nanotubes subjected to a longitudinal magnetic field and a hygrothermal environment. The results showed that a longitudinal magnetic field could increase critical flow velocities and natural frequencies of the single-walled carbon nanotubes.

All of the above studies were aimed at functionally graded fluid-conveying nanotubes, and there have been relatively few studies on the stability of micrometer-scale functionally graded fluid-conveying microtubes. For example, Babaei et al. [29] derived the nonlinear equation of motion of functionally graded, porous, curved microtubes based on higher-order shear deformation tube theory and Hamilton’s principle. A two-step perturbation technique was used to solve the nonlinear equations of motion, and the effects of the material length scale parameters, functional grading mode, porosity parameters, and nonlinear elastic foundation on the system stability were analyzed. She et al. [30] established the governing equation of functionally graded porous tubes via a nonlocal strain gradient theory and analyzed the nonlinear bending and vibrational characteristics of porous microtubes. Ansari et al. [31] combined modified coupled stress theory with first-order shear deformation shell theory and deduced the motion equation and boundary conditions of the system using Hamilton’s principle. The results demonstrated that the stability of the microtube could be improved by increasing the value of the material property gradient index. Talib et al. [32,33] studied the influence of the fluid flow velocity, gradient index, and parameters of the material length scale on the vibrations and stability of functionally graded, fluid-conveying microtubes. Setoodeh et al. [34] derived the governing equation based on Euler–Bernoulli beam theory, strain gradient theory, and von Kármán geometric nonlinearities, and they used an analytical method to determine the size-dependent nonlinear vibrational behaviors of functionally graded fluid-conveying microtubules.

To better understand the vibration behavior of functionally graded fluid-conveying microtubules, based on strain gradient theory and Hamilton’s principle, the governing equation of functionally graded simply supported fluid-conveying microtubes under multiple physical fields is established in this paper. The properties of functionally graded materials showed a power law distribution along the fluid-conveying microtube radius. The Galerkin mothed was used to solve the governing equations, and the effects of the microscale parameters, temperature change, and magnetic field on the vibration behavior of the system are discussed in detail.

## 2. Governing Equations

In the present study, the schematic of a functionally graded simply supported fluid-conveying microtube subjected to multiple physical fields is shown in Figure 1. In this schematic, the length of the microtube is L, the velocity of the fluid is U, the inner radius of the microtube is $R_{i}$, the outer radius of the microtube is $R_{o}$, the radius of the reference point is r, the mass of a unit length of the microtube is m, and the fluid mass of a unit length of the microtube is M.

A functionally graded material is an advanced composite material, whose mechanical properties are continuously changing. In this study, it is assumed that the mechanical properties of the functionally graded material change continuously across the microtube wall thickness according to a power-law function. The mechanical properties of the functionally graded material can be represented as [35]
(1)$$E\left(r\right)=V_{i}E_{i}+V_{o}E_{o}$$
(2)$$G\left(r\right)=V_{i}G_{i}+V_{o}G_{o}$$
(3)$$\rho \left(r\right)=V_{i}\rho_{i}+V_{o}\rho_{o}$$
(4)$$V_{i}={\left(\frac{R_{o}-r}{R_{o}-R_{i}}\right)}^{n}$$
(5)$$V_{o}=1-V_{i}$$
where E, G, and ρ denote the elastic modulus, shear modulus, and density of the functionally graded material, respectively. Subscripts i and o denote the inner and outer surfaces, respectively. V is the material volume fraction, and superscript n represents the power-law index. In this work, the inner layer material of the functionally graded simply supported fluid-conveying microtube was a zirconia ceramic, and the outer layer was aluminum. Their material properties were $E_{i}=151\;\mathrm{GPa}$, $\rho_{i}=5331{\;\mathrm{kg}/m}^{3}$, $a_{i}=9.5\times 10^{-6}{\;K}^{-1}$, $E_{o}=70\;\mathrm{GPa}$, $\rho_{o}=2707{\;\mathrm{kg}/m}^{3}$, $a_{o}=23.6\times 10^{-6}{\;K}^{-1}$.

From Equations (1)–(5), it is known that when the index n=0, the functionally graded material degenerates into a single homogeneous ceramic material. When the index n=∞, the functionally graded material degenerates into a single homogeneous metal material.

Strain gradient theory is a widely used high-order continuum mechanics theory that has been derived in the literature [13]. According to strain gradient theory, the strain energy of an isotropic linear elastic continuum with a small deformation can be written as
(6)$$U_{p}=\frac{1}{2}\int_{0}^{L}\left[S{\left(\frac{\partial^{2}w}{\partial x^{2}}\right)}^{2}+K{\left(\frac{\partial^{3}w}{\partial x^{3}}\right)}^{2}\right]dx$$
where $S=EI+2GAl_{0}^{2}+\frac{8}{15}GAl_{1}^{2}+GAl_{2}^{2}$, and $K=2GIl_{0}^{2}+\frac{4}{5}GIl_{1}^{2}$. The parameters w, $l_{i}\left(i=0,1,2\right)$, I, and A are the transverse displacement of the microtube, length scale parameters of the functionally graded material, cross-sectional moment of inertia of the microtube, and the cross-sectional area of the microtube, respectively.

The kinetic energy of a microtube can be expressed as
(7)$$T_{p}=\frac{m}{2}\int_{0}^{L}{\left(\frac{\partial w}{\partial t}\right)}^{2}dx$$

The kinetic energy of the fluid in the microtube can be expressed as
(8)$$T_{f}=\frac{M}{2}\int_{0}^{L}\left[{\left(\frac{\partial w}{\partial t}+U\frac{\partial w}{\partial x}\right)}^{2}+U^{2}\right]dx$$

According to Maxwell’s equation, the work of a Lorentz force due to a longitudinal magnetic field acting on the microtube can be expressed as [27]
(9)$$W_{m}=\sigma AH_{x}^{2}\int_{0}^{L}\frac{\partial^{2}w}{\partial x^{2}}dx$$
where $H_{x}$ and σ are the magnetic field intensity and permeability, respectively.

According to thermoelastic theory, the thermal axial force generated by a temperature difference and the work done by the thermal axial force on the microtube can be written as [22]
(10)$$W_{T}=-\int_{0}^{L}\frac{EA}{1-2\nu }a\Delta T\frac{\partial^{2}w}{\partial x^{2}}dx$$
where a, ν, and ΔT are the coefficient of thermal expansion, Poisson’s ratio, and the temperature difference of the temperature field, respectively.

The governing equation of the microtube can be obtained by applying Hamilton’s principle as follows:(11)$$\delta \int_{t_{1}}^{t_{2}}\left(T_{p}+T_{f}-U_{p}+W_{ext}-MU^{2}u_{L}\right)dt=\int_{t_{1}}^{t_{2}}MU\left(\frac{\partial w_{L}}{\partial t}+U\frac{\partial w_{L}}{\partial x}\right)\delta w_{L}dt$$
where $W_{ext}=W_{m}+W_{T}$. At both ends of a simply supported fluid-conveying microtube, $u_{L}=w_{L}=0$, and therefore, Equation (11) can be simplified as
(12)$$\delta \int_{t_{1}}^{t_{2}}\left(T_{p}+T_{f}-U_{p}+W_{ext}\right)dt=0$$

By substituting Equations (6)–(10) into Equation (12), the governing equation of a simply supported fluid-conveying microtube can be obtained as follows:(13)$$S\frac{\partial^{4}w}{\partial x^{4}}-K\frac{\partial^{6}w}{\partial x^{6}}+\left(m+M\right)\frac{\partial^{2}w}{\partial t^{2}}+2MU\frac{\partial^{2}w}{\partial t\partial x}+\left(MU^{2}-\sigma AH_{x}^{2}+\frac{EA}{1-2\nu }a\Delta T\right)\frac{\partial^{2}w}{\partial x^{2}}=0$$

For convenience of calculation, we introduce the following dimensionless quantities:$$\begin{array}{c} \xi =\frac{x}{L},\eta =\frac{w}{L},\;\tau =\frac{t}{L^{2}}\sqrt{\frac{EI}{m+M}},\;u=\sqrt{\frac{M}{EI}}UL,\;\beta =\frac{M}{m+M},\;\varpi =\sigma \frac{AH_{x}^{2}L^{2}}{EI},\;\phi =\frac{(EA)L^{2}a\Delta T}{(EI)\left(1-2v\right)},\\ \varphi =\frac{GIl_{2}^{2}}{EIL^{2}},\lambda =\frac{GAl_{2}^{2}}{EI},\;\mu =\frac{EI}{GAl_{2}^{2}},\;\psi =\lambda \left(\mu +2{r_{0}}^{2}+\frac{8}{15}{r_{1}}^{2}+1\right),\;\kappa =2\varphi \left({r_{0}}^{2}+\frac{2}{5}{r_{1}}^{2}\right),\;r_{0}=\frac{l_{0}}{l_{2}},\;r_{1}=\frac{l_{1}}{l_{2}} \end{array}$$

By substituting the dimensionless parameters above into Equation (13), the governing equation of the simply supported fluid-conveying microtube can be written as
(14)$$\frac{\partial^{2}\eta }{\partial \tau^{2}}+2u\sqrt{\beta }\frac{\partial^{2}\eta }{\partial \xi \partial \tau }+\psi \frac{\partial^{4}\eta }{\partial \xi^{4}}+\left(u^{2}-\varpi +\phi \right)\frac{\partial^{2}\eta }{\partial \xi^{2}}-\kappa \frac{\partial^{6}\eta }{\partial \xi^{6}}=0$$

For a simply supported fluid-conveying microtube, the boundary conditions are as follows:(15)$$\begin{array}{l} w(0)=w(L)=0,u(0)=u(L)=0,w^{\prime \prime \prime }(0)=w^{\prime \prime \prime }(L)=0\\ S\cdot w^{\prime \prime }(0)-K\cdot w^{⁗}(0)=S\cdot w^{\prime \prime }(L)-K\cdot w^{⁗}(L)=0 \end{array}$$

## 3. Galerkin Method

Equation (14) is a high-order partial differential equation, which can be discretized into a second-order ordinary differential equation by the Galerkin method. To this end, the transverse vibration displacement of the microtube is expressed as follows:(16)$$\eta (\xi ,\tau )=\sum_{r=1}^{N}\varphi_{r}(\xi )q_{r}(\tau )$$
where N is the order of the mode, $q_{r}\left(\tau \right)$ is the *r*-th mode coordinate, and $\varphi_{r}(\xi )$ is the basis function of the *r*-th eigenmode. For a simply supported fluid-conveying microtube, the basis function of the *r*-th eigenmode is shown as follows:(17)$$\varphi_{r}(\xi )=\sqrt{2}\sin(\lambda_{r}\xi )$$
where $\lambda_{r}$ is the *r*-th-order eigenvalue of the corresponding simply supported beam.

By substituting Equations (16) and (17) into Equation (14), multiplying both sides of the equation by $\varphi_{s}(\xi )$, integrating with respect to ξ over the interval [0,1], and making use of the orthogonal property of the characteristic function, the following equation can be obtained:(18)$$\begin{array}{r} \sum_{r=1}^{N}\varphi_{s}(\xi )\varphi_{r}(\xi ){\ddot{q}}_{r}(\tau )+\sum_{r=1}^{N}\left(u^{2}-\varpi +\phi \right)\varphi_{s}(\xi )\varphi_{r}^{\prime \prime }(\xi )q_{r}(\tau )+\sum_{r=1}^{N}2u\sqrt{\beta }\varphi_{s}(\xi )\varphi_{r}^{\prime }(\xi ){\dot{q}}_{r}(\tau )\\ +\sum_{r=1}^{N}\left(1+\psi \right)\lambda_{r}^{4}\varphi_{s}(\xi )\varphi_{r}(\xi )q_{r}(\tau )-\sum_{r=1}^{N}\kappa \varphi_{s}(\xi )\varphi_{r}^{\left(6\right)}(\xi )q_{r}(\tau )=0 \end{array}$$

Equation (18) can be written in matrix form as follows:(19)$$M{\ddot{q}}_{r}(\tau )+C{\dot{q}}_{r}(\tau )+Kq_{r}(\tau )=0$$
where $q_{r}(\tau )$ is the population vector in generalized coordinates, and (⋅) is the derivative with respect to time τ. The mass matrix [M], the damping matrix [C], and the stiffness matrix [K] can be written as
$${\left[M\right]}_{sr}=\int_{0}^{1}\varphi_{s}(\xi )\varphi_{r}(\xi )d\xi ;\;\;\;\;\;\;{\left[C\right]}_{sr}=2u\sqrt{\beta }\int_{0}^{1}\varphi_{s}(\xi )\varphi_{r}^{\prime }(\xi )d\xi ;$$
$${\left[K\right]}_{sr}=(u^{2}-\varpi +\phi )\int_{0}^{1}\varphi_{s}(\xi )\varphi_{r}^{\prime \prime }(\xi )d\xi +\psi \lambda_{r}^{4}\int_{0}^{1}\varphi_{s}(\xi )\varphi_{r}(\xi )d\xi -\kappa \int_{0}^{1}\varphi_{s}(\xi )\varphi_{r}^{\left(6\right)}(\xi )d\xi$$

Equation (19) can be converted into a first-order equation of state, and all modes of the system can be solved using the MATLAB software. The stability characteristics of the system can be judged by the eigenvalues.

In the expression of the present solution, the eigenfrequency is a complex value, the real part of the eigenfrequency represents the damping of the system, and the imaginary part of the eigenfrequency represents the frequency of the system. If the real part Re(ω)=0 and the imaginary part Im(ω)>0, the system is stable. If the real part Re(ω)<0 and the imaginary part Im(ω)=0, the system loses its stability by divergence. If the real part Re(ω)<0 and the imaginary part Im(ω)>0, the flutter instability occurs.

## 4. Numerical Results and Discussion

In this section, the length scale parameters of functionally graded materials were selected as $l=l_{0}=l_{1}=l_{2}=17.6\;\mu m$ [36], the ratio of the inner and outer diameters was *d*/*D* = 0.8, the ratio of the length to the outer diameter was L/D=20, the density of the fluid in the microtube was $\rho_{f}=1000{\;\mathrm{kg}/m}^{3}$, the magnetic permeability was $\sigma =4\pi \times 10^{-7}\;H/m$, and the dimensionless microscale parameter was δ=D/l. We will investigate the effects of the dimensionless microscale parameters, temperature differences, and magnetic field intensity on the dynamic behaviors of the microtube.

### 4.1. Validation of Model

To verify the correctness of the calculation results, the material length scale parameters were ignored, and the calculated results from this work and the literature [37] are plotted in Figure 2 for the same parameter values. The results of both calculations were in good agreement, which proved the correctness of the results from this work.

### 4.2. Influence of Parameters on Vibration Characteristics of Microtube

In the following study, the dimensionless microscale parameter values of δ=1 and 3 and the magnetic field intensity values of $H_{x}=0$, $2\times 10^{7}$, and $4\times 10^{7}\;A/m$ were considered. ΔT is the temperature difference relative to room temperature, and its values were ΔT=0, 50, and 100 K. The stability characteristics of the microtube under different parameters were studied.

First, the magnetic field effects were ignored, and the dimensionless microscale parameter was set to δ=1 and 3. The stability of the microtube at various dimensionless velocities and three different temperature differences for δ=1 was determined, as shown in Figure 3.

It can be seen from Figure 3 that with the increase in the dimensionless velocity, the frequency of the system gradually decreased. When the first-order frequency was zero, the first-order damping bifurcated, and the system diverged. The corresponding dimensionless flow velocity is called the critical velocity. The critical velocity of the system at the temperature differences of 100, 50, and 0 K were 10.683, 11.131, and 11.654, respectively. When the dimensionless flow velocity was zero, the corresponding first-order natural frequencies were 36.6088, 34.9673, and 33.2447, and the second-order natural frequencies were 146.8286, 145.2193, and 143.592, respectively.

Figure 4 presents the dimensionless eigenfrequency variations with the dimensionless flow velocity for three temperature differences when the dimensionless microscale parameter was δ=3. When the first-order frequency was zero, the critical velocities of the divergence of the microtube were 1.166, 3.645, and 5.018, respectively. However, when the dimensionless flow velocity was 8.792, 9.441, and 10.041, respectively, the first-order frequency increased from zero again and coupled with the second-order mode, resulting in the mode-coupled flutter instability of the microtube. In addition, when the dimensionless flow velocity was 14.327, 14.733, and 15.122, respectively, the first-order mode diverged again, and the second-order and third-order modes underwent coupled flutter instability. When the dimensionless velocity was zero, the first-order natural frequencies were 3.662, 11.441, and 15.761, and the second-order natural frequencies were 55.202, 59.307, and 63.149, respectively.

It can be seen from Figure 3 and Figure 4 that when with the increase in the temperature difference, the critical velocity and natural frequency of the microtube decreased, and the microtube was more prone to instability. When the temperature difference was the same, the larger the dimensionless microscale parameter was, the smaller the critical velocity and natural frequency of the microtube became. Moreover, with the increase in the dimensionless velocity, the divergent instability of the microtube occurred first, and then the mode-coupled flutter instability occurred. The reason for this change was that the larger the dimensionless microscale parameter was, the larger the microtube structure size was, and the weaker the microscale effect of the material became.

Next, the temperature effects were ignored, and dimensionless microscale parameters of δ=1 and 3 and magnetic field intensities of $H_{x}=0$, $2\times 10^{7}$, and $4\times 10^{7}\;A/m$ were selected. The stability of the microtube under different values of the magnetic field intensity was analyzed, as shown in Figure 5 and Figure 6.

Figure 5 shows the dimensionless eigenfrequency variations with the dimensionless flow velocity under different magnetic fields when the dimensionless microscale parameter was δ=1. The microtube frequency gradually decreased as the dimensionless flow velocity increased. When the first-order frequency decreased to zero, the critical velocity of the microtube under magnetic field strengths of $H_{x}=0$, $2\times 10^{7}$, and $4\times 10^{7}\;A/m$ were 11.653, 12.799, and 15.745, respectively. In addition, when the dimensionless flow velocity u=0, the first-order natural frequencies were 36.609, 40.209, and 49.462, and the second-order natural frequencies were 146.827, 150.549, and 161.195, respectively.

Figure 6 shows the dimensionless eigenfrequency variations with the dimensionless flow velocity under different magnetic fields when the dimensionless microscale parameter was δ=3. The critical velocity of the microtube at magnetic field strengths of $H_{x}=0$, $2\times 10^{7}$, and $4\times 10^{7}\;A/m$ were 5.017, 7.294, and 11.716, respectively. When the dimensionless flow velocity increased to 10.039, 11.716, and 14.486, the first-order mode frequency increased from zero, resulting in mode-coupled flutter with the second-order modal coupling. Moreover, with the increase in the dimensionless flow velocity, the first-order mode diverged again, the second and third modes were coupled, and flutter instability occurred. When the dimensionless flow velocity was u=0, the first-order natural frequencies corresponding to magnetic field strengths of $H_{x}=0$, $2\times 10^{7}$, and $4\times 10^{7}\;A/m$ were 15.761, 22.912, and 36.806, and the second-order natural frequencies were 63.154, 41.369, and 91.719, respectively.

It can be concluded from Figure 5 and Figure 6 that increasing the magnetic field intensity could significantly improve the critical velocity and stability of the microtube. When the magnetic field intensity was the same, the larger the dimensionless microscale parameter was, the smaller the critical velocity was, and the worse the stability of the microtube became.

In the work described above, the stability of functionally graded, simply supported, fluid-conveying microtubes with a single physical field varied was conducted, but the stability of the system under multi-field coupling was not considered. Therefore, temperature differences of ΔT=0 and 50 K and magnetic field intensities of $H_{x}=0$ and $2\times 10^{7}\;A/m$ were selected, and the stability of the microtube in the thermo-magnetic coupled field was studied. The results are shown in Figure 7 and Figure 8.

As shown in Figure 7, when ΔT=0 K and $H_{x}=0\;A/m$, the critical velocity of the microtube was $u_{cr}=11.653$. When ΔT=50 K and $H_{x}=0\;A/m$, the critical velocity of the microtube was $u_{cr}=11.132$. When ΔT=0 K and $H_{x}=2\times 10^{7}\;A/m$, the critical velocity of the microtube was $u_{cr}=12.799$. When ΔT=50 K and $H_{x}=2\times 10^{7}\;A/m$, the critical velocity of the microtube was $u_{cr}=12.326$. Owing to the opposite effects of the temperature difference and magnetic field, the critical velocity of the microtube in the coupled field was between those of the two single physical fields.

Similar results can be found in Figure 8. When ΔT=0 K and $H_{x}=0\;A/m$, the critical velocity of the microtube was $u_{cr}=5.017$. When ΔT=50 K and $H_{x}=0\;A/m$, the critical velocity of the microtube was $u_{cr}=3.642$. When ΔT=0 K and $H_{x}=2\times 10^{7}\;A/m$, the critical velocity of the microtube was $u_{cr}=7.294$. When ΔT=50 K and $H_{x}=2\times 10^{7}\;A/m$, the critical velocity of the microtube was $u_{cr}=6.426$. In addition, due to the increase in the dimensionless micro-scale parameters, the microscale effect of the material decreased. For the four cases, when the dimensionless flow velocity increased to 9.44, 10.141, 10.784, and 11.323, respectively, the first- and the second-order modes of the microtube were coupled, leading to flutter instability. It can be concluded from Figure 7 and Figure 8 that the stability of the microtube under multiple physical fields depends on the parameter values of each physical field. The magnetic field can increase the stability of the microtube, while the temperature field can reduce the stability of the microtube.

### 4.3. Influence of Parameters on Critical Velocity of Microtube

Figure 9 shows the variations in the critical velocity of the microtube with the temperature difference and magnetic field strength. With the increase in the temperature difference, the critical velocity of the microtube decreased, while the critical velocity of the microtube increased with the increase in the magnetic field intensity. Furthermore, the smaller the dimensionless microscale was, the greater the critical velocity and the more stable the microtubes were. When the dimensionless microscale parameter was δ=3, the microtube was prone to divergence at low flow velocities and flutter instability at high flow velocities. However, when the temperature difference was ΔT=45 K, the critical velocity of the divergence was reduced to zero. Therefore, controlling the temperature difference and increasing the magnetic field intensity can effectively improve the critical velocity and stability of the microtube.

## 5. Conclusions

In this paper, we investigated the stability of functionally graded simply supported fluid-conveying microtubes under multi-physical fields. The Galerkin method was used to solve the eigenfrequency of the microtube, and the effects of temperature change and magnetic field intensity on the stability of the microtube at different dimensionless microscales were analyzed. As well, some conclusions were obtained as follows:

Increasing the dimensionless microscale parameter will lead to a decrease in the eigenfrequency and critical velocity. When the dimensionless microscale parameter was δ=3, with the increase in the dimensionless flow velocity, the instability of the microtube changed from first-order mode divergence to a higher-order mode-coupled flutter instability, which lowered the stability of the microtube.

Increasing the temperature difference will lead to decreases in the eigenfrequency and critical velocity, and the greater the temperature difference is, the worse the stability of the microtube will become. Under the action of a magnetic field, with a higher magnetic field intensity, the natural frequency and critical velocity of the microtube will increase, and the microtube will become more stable.

The stability of the microtube can be significantly improved by decreasing the temperature difference and increasing the magnetic field intensity under the action of a thermomagnetic coupled field. Therefore, in a high-temperature field, the influence of the temperature difference on the stability of the system can be effectively reduced by selecting an appropriate magnetic field intensity to improve the stability of the microtube.

## Figures and Tables

**Figure 1 micromachines-13-00895-f001:**
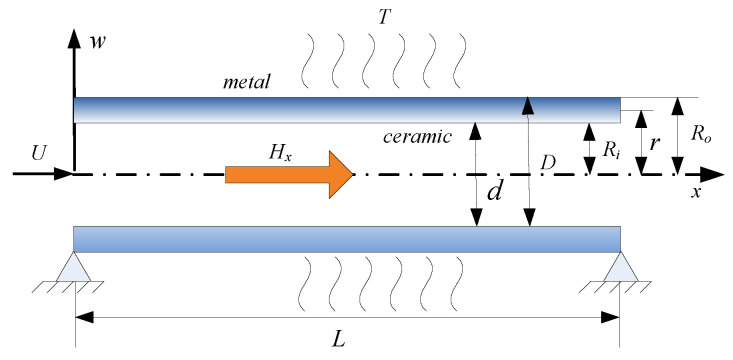
Schematic of the functionally graded, simply support, fluid-conveying microtube under multiple physical fields.

**Figure 2 micromachines-13-00895-f002:**
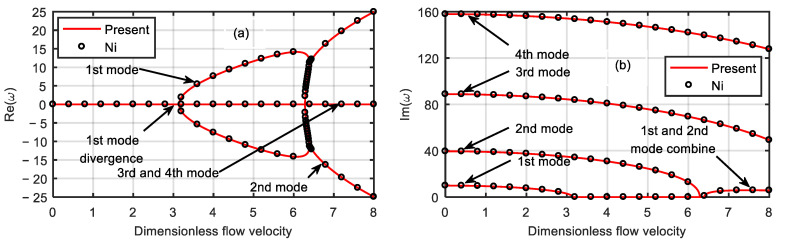
Variations of first four order eigenfrequencies with the dimensionless flow velocity. (**a**) Real part. (**b**) Imaginary part.

**Figure 3 micromachines-13-00895-f003:**
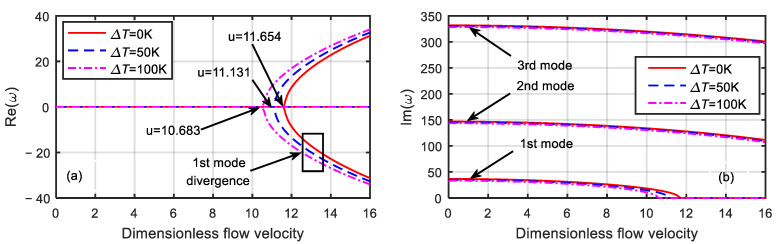
Dimensionless eigenfrequency variations with dimensionless flow velocity for various temperature differences (δ=1). (**a**) Real part. (**b**) Imaginary part.

**Figure 4 micromachines-13-00895-f004:**
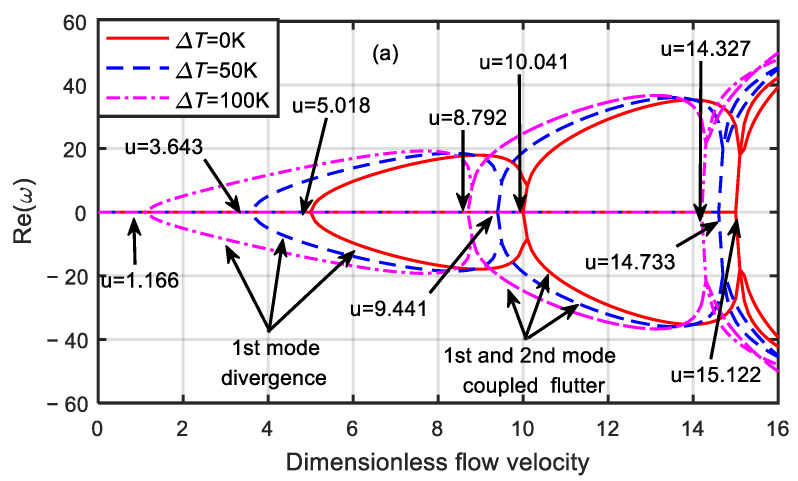
Dimensionless eigenfrequency variations with dimensionless flow velocity for various temperature differences (δ=3). (**a**) Real part. (**b**) Imaginary part.

**Figure 5 micromachines-13-00895-f005:**
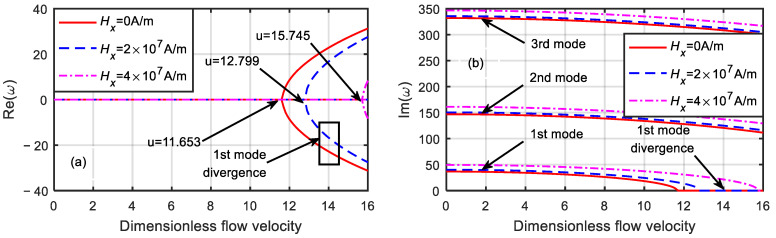
Dimensionless eigenfrequency variations with dimensionless flow velocity under different magnetic fields (δ=1). (**a**) Real part. (**b**) Imaginary part.

**Figure 6 micromachines-13-00895-f006:**
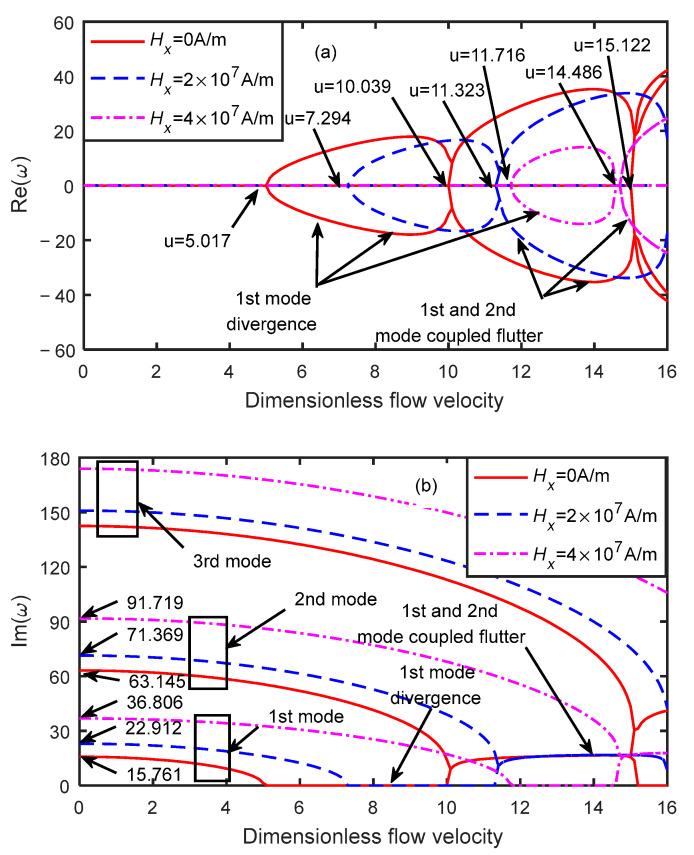
Dimensionless eigenfrequency variations with dimensionless flow velocity under different magnetic fields (δ=3 ). (**a**) Real part. (**b**) Imaginary part.

**Figure 7 micromachines-13-00895-f007:**
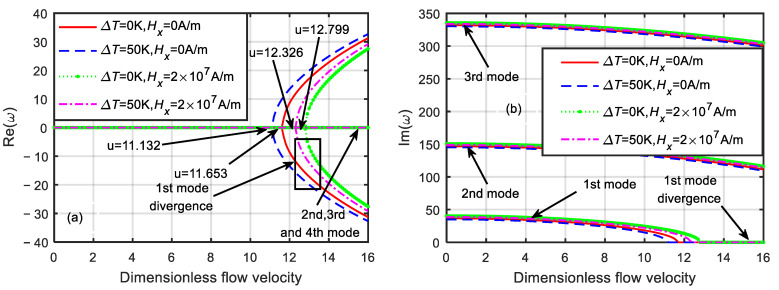
Dimensionless eigenfrequency variations with dimensionless flow velocity under different thermo-magnetic coupled fields (δ=1). (**a**) Real part. (**b**) Imaginary part.

**Figure 8 micromachines-13-00895-f008:**
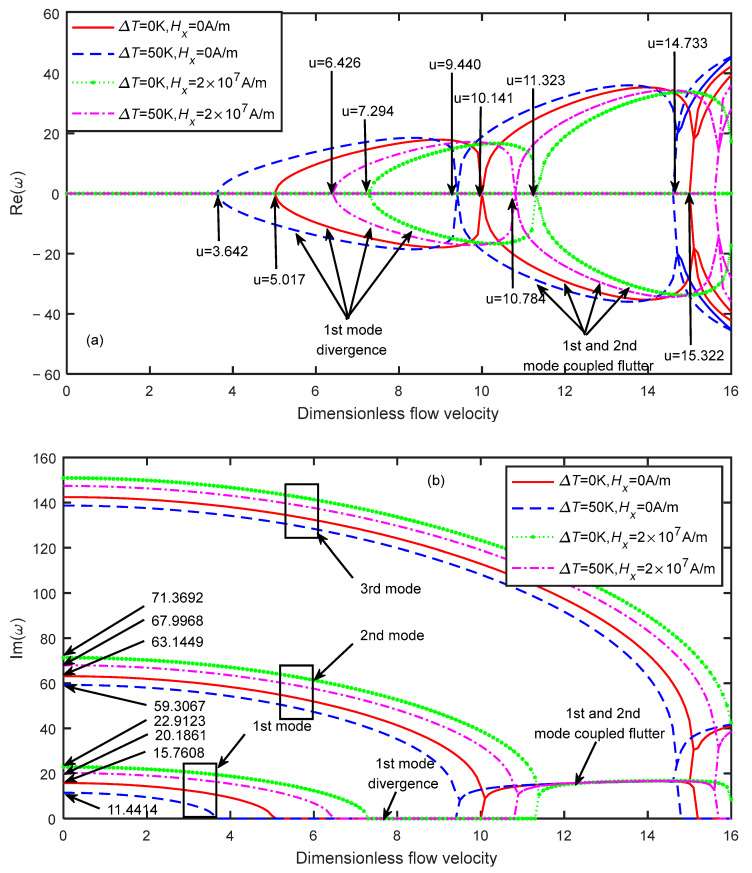
Dimensionless eigenfrequency variations with dimensionless flow velocity under different thermo-magnetic coupled fields (δ=3). (**a**) Real part. (**b**) Imaginary part.

**Figure 9 micromachines-13-00895-f009:**
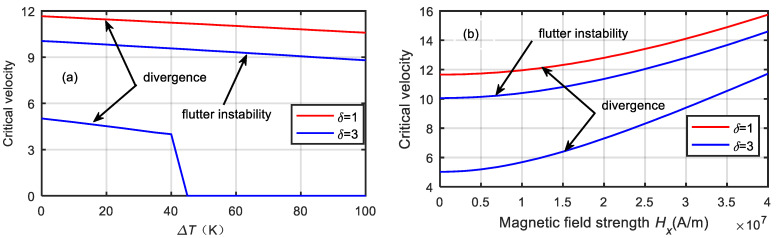
Critical velocity variations of microtube with temperature difference and magnetic field strength. (**a**) temperature difference. (**b**) magnetic field strength.

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
