# Peer review of "Study on the Stability of Functionally Graded Simply Supported Fluid-Conveying Microtube under Multi-Physical Fields"

_micromachines, 2022, doi:10.3390/mi13060895_

Round 1

Reviewer 1 Report

The manuscript studies the stability of a simply supported fluid-conveying microtube under the influence of a magnetic field. The microtube is considered as a two-dimensional channel where the mechanical properties of the wall changing continuously across the thickness of wall. The fluid flow is considered to be incompressible, and the governing equations are solved using the Galerkin method. Authors investigate the stability of the microtube while changing the parameters.

In my view the manuscript can be considered for publication in Micromachines after addressing the following comments:

  • Some of the equations in the manuscript do not align with the text, for example please check the paragraphs below equation (6), (9), (10), (11).
  • Authors should explain all the parameters used in the equations, for example \sigma in equation (9).
  • Authors nondimensionalize the governing equations in section two, however the effect of the dimensional parameters is studied later in the manuscript. It will provide useful information if authors can show the effect of each dimensionless parameter on the stability, for example instead \phi instead of \Delta T.

Author Response

Point 1: Some of the equations in the manuscript do not align with the text, for example please check the paragraphs below equation (6), (9), (10), (11)。

Response 1: Thank you very much for your comments. According to your comments, we have modified all the equations in the manuscript do not align with the text and marked them in red in the article.

Point 2: Authors should explain all the parameters used in the equations, for example \sigma in equation (9)

Response 2: We are very sorry for our incorrect writing. The sigma () was wrongly written as rho () at the paragraphs below Eq.(9), which has been corrected to sigma () in the article and marked in red. In addition, all equation writing was checked to ensure that all parameters were explained. The symbols including those in Figure 1 have also been modified.

Point 3:  Authors nondimensionalize the governing equations in section two, however the effect of the dimensional parameters is studied later in the manuscript. It will provide useful information if authors can show the effect of each dimensionless parameter on the stability, for example instead \phi instead of \Delta T.

Response 3: Thank you very much for your suggestion. For the convenience of calculation, Eq. (13) was dimensionless. However, when studying the effects of temperature and magnetic field on the stability of the fluid-conveying microtube system, we take temperature change( \Delta T) and magnetic fields( \)as the main variables to study its effect on the stability of the fluid-conveying microtube system, and then do not analyze every dimensionless parameter in this article.

Reviewer 2 Report

The authors comprehensively analysed the stability of functionally graded simply supported fluid-conveying microtubes under 10 multiple physical. They did a good job as presented in this manuscript. I have a few minor comments to make:

1. In using the eigenvalues to determine the stability of the system, could the authors ensure all possible combination of imaginary and real parts are covered. It would also be useful if this result is presented in a tabular form for ease of assimilation

2. Could the authors include some quantitative results in the conclusion and abstract. Using a case study, can the authors quantify the effect of the magnetic fields on microchannel stability?

3. Could the authors clearly state how current work differs from previous work of Ansari [31], Talib [32, 33] etc. Where applicable, a comparative analysis of current studies with results from present study on a case example could proof useful.

Author Response

Point 1: In using the eigenvalues to determine the stability of the system, could the authors ensure all possible combination of imaginary and real parts are covered. It would also be useful if this result is presented in a tabular form for ease of assimilation

Response 1: It is a good suggestion to present the eigenvalues in tabular form, However, this paper adopts the numerical analysis method, and the number of eigenvalues is very large. If the eigenvalues are listed in a table, the table will be very large.

Point 2: Could the authors include some quantitative results in the conclusion and abstract. Using a case study, can the authors quantify the effect of the magnetic fields on microchannel stability?

Response 2: Thank you for this valuable feedback. These quantitative results can be put in the abstract and conclusion, but due to the large number of quantitative analysis data, we put the quantitative analysis results in the fourth part of the paper. In view of the influence of magnetic field on the stability of the fluid-conveying microtubule, magnetic field intensities of , , and  were selected to study their influence on the stability.

Point 3: Could the authors clearly state how current work differs from previous work of Ansari [31], Talib [32, 33] etc. Where applicable, a comparative analysis of current studies with results from present study on a case example could proof useful.

Response 3: Ref [31] investigated the effects of length-scale parameters, material property gradient exponents, compressive axial loads, and long-to-radius ratios on the stability of the fluid-conveying microtube. Ref [32,33] used analytical and semi-analytical methods to solve the vibration equations, respectively, and studied the influence of the fluid flow velocity, gradient index, and parameters of the material length scale on the vibrations and stability of functionally graded, fluid-conveying microtubes. In our paper, we used numerical analysis method to solve the motion equation and studied the influence of temperature and magnetic field intensity changes on the stability of the system under different microscale parameters.

Due to the limitation of the conditions, we are not able to compare and analyze current studies with specific experimental results, which is also our biggest issue. 

Reviewer 3 Report

Thank you for your invitation to review your valuable research achievement.  The manuscript was well written and also results are well presented. Only I want to recommend technical points of your manuscript.

- Please define your symbols in equations, I recommend nomeclature as a separate part.

-In your figures, real and imaginary part were distinguished from (a) and (b). But I recommend to indicate (a) and (b) in the caption of each plot.

-In your validation of Fig. 2, validation must be verified by experimental results or any conidential method. But Ni's result is also numerical result. Is it valuable for validation?

- As described in Ref [38], your model also based on Cantilever. Then I recommend to present your calculation model, not just a pipe in Fig. 1.

Author Response

Point 1: Please define your symbols in equations, I recommend nomeclature as a separate part.

Response 1: Thank you very much for your comments. According to your comments, we have checked all the symbols in the equation and explained them at the end of the equation.

Point 2: In your figures, real and imaginary part were distinguished from (a) and (b). But I recommend to indicate (a) and (b) in the caption of each plot.

Response 2: We are very sorry for our negligence in the picture. According to your comments, we have indicated (a) and (b) in the caption of each picture.

Point 3: In your validation of Fig. 2, validation must be verified by experimental results or any conidential method. But Ni's result is also numerical result. Is it valuable for validation?

Response 3: Thank you for this valuable suggestion. Due to the limitation of the conditions, we cannot complete the experimental verification of the fluid-conveying microtube at present, which is also our biggest issue. According to our literature review, there is no experimental data that can be used for comparative verification. In order to ensure the credibility of the numerical analysis in this paper, the correctness of the equation derivation in this paper can be verified by comparing with the Ni’s result, which indirectly proves the correctness of the numerical analysis in this paper.

Point 4: As described in Ref [38], your model also based on Cantilever. Then I recommend to present your calculation model, not just a pipe in Fig. 1.

Response 4: Thank you for that excellent and insightful series of remarks. Ref [38] established four different models, including cantilevered pipe, pinned-pinned pipe, clamped-clamped pipe and clamped-clamped pipe, and studied the application of the differential transformation method in the vibration analysis of pipes conveying fluid. In ours paper, the motion equation was established based on the pinned-pinned pipe model, as shown in Fig.1. The microscale effect was ignored and the calculated results were compared with the results of the pinned-pinned pipe in Ref [38] to verify the correctness of equation derivation in this paper.
